# Ultrafast all-optical toggle writing of magnetic bits without relying on heat

T. Zalewski [1], A. Maziewski[1], A. V. Kimel[2] & A. Stupakiewicz [1] ✉

Ultrafast excitation of matter can violate Curie's principle that the symmetry of the cause must be found in the symmetry of the effect. For instance, heating alone cannot result in a deterministic reversal of magnetization. However, if the heating is ultrafast, it facilitates toggle switching of magnetization between stable bit-states without any magnetic field. Here we show that the regime of ultrafast toggle switching can be also realized via a mechanism without relying on heat. Ultrafast laser excitation of iron-garnet with linearly polarized light modifies magnetic anisotropy and thus causes toggling magnetization between two stable bit states. This new regime of 'cold' toggle switching can be observed in ferrimagnets without a compensation point and over an exceptionally broad temperature range. The control of magnetic anisotropy required for the toggle switching exhibits reduced dissipation compared to laser-induced-heating mechanism, however the dissipation and the switching-time are shown to be competing parameters.

The field of ultrafast magnetism studies the processes of magnetization dynamics triggered by stimuli much shorter than the characteristic time required to reach thermodynamic equilibrium in magnets[1,2]. The field was started with the seminal discovery of femtosecond-laser-induced demagnetization of ferromagnetic Ni on a timescale much faster than the characteristic time of any known at that time elementary interaction of spins with other reservoirs of angular momentum in solids[3]. Afterward, the interest in the field was fueled by a plethora of counter-intuitive experimental findings that challenged state-of-the-art theories in magnetism[4–7]. Moreover, it is also believed that ultrafast magnetism can help us to understand the fundamental limits on the time as well as the energy dissipation required for writing magnetic bits[8]. Thus, the field has a potential for a tremendous impact on future magnetic data storage technology. Special attention in this case is paid to materials, mechanisms, and scenarios to switch magnetization between stable bit states at the fastest possible and, simultaneously, the least dissipative way. Until now the absolute majority of studies in ultrafast magnetism have been dedicated to metallic magnets. Laser-matter interaction in the case of metals is dominated by the interaction of the electric field of light with free electrons, which unavoidably results in irreversible energy transfer from light to matter, entropy increase, and dissipations.

For instance, the discovery of ultrafast all-optical toggle switching magnetization in metallic ferrimagnetic alloy GdFeCo[9] is probably one of the most impactful in the field. It inspired the discovery of current-induced toggle switching in GdFeCo[10,11] and studies of the switching mechanisms in other classes of metallic ferrimagnets[12–14]. In all these cases, ultrafast heating of the ferrimagnet on a timescale much faster than the characteristic time scale of interaction between the spins of two magnetic sublattices was one of the key requirements for the realization of the mechanism[15]. The ultrafast heating was shown to turn the ferrimagnets into such a strongly non-equilibrium state that the preferable route for the subsequent relaxation from this state has to be accompanied by magnetization reversal. The laser-induced toggle switching without relying on heat would tremendously decrease dissipations during the writing of magnetic bits, but so far seemed to be unrealistic.

Aiming to find media for the "cold" toggle switching, we would like to emphasize that such a switching, in principle, contradicts thermodynamic Curie's principle[16], according to which symmetries of the causes are to be found in the symmetry of the effect[17]. Indeed, while every excitation event is identical and thus the symmetry of the cause does not change, toggling implies that the symmetry of the effect alters from pulse to pulse. Such a toggle switching becomes possible,

[1]Faculty of Physics, University of Bialystok, Bialystok, Poland. [2]Radboud University, Institute for Molecules and Materials, Nijmegen, The Netherlands.
✉e-mail: and@uwb.edu.pl

for instance, in the case of ultrafast precessional switching, when a pulse of a magnetic field is applied perpendicular to the magnetization precisely during the half of the precession period[18–20]. Thermodynamic Curie's principle in this case is no longer applicable and a match of cause-response symmetries is no longer required, because the field is applied on a time scale, which is much shorter than the time required for the magnetization to arrive at the thermodynamic equilibrium, i.e., become parallel with respect to the applied magnetic field. Hence, to demonstrate "cold" all-optical toggle switching, one has to find a way to generate an effective magnetic field acting on the magnetization on a time scale of half the period of the magnetization precession.

In order to generate such effective fields and demonstrate "cold" all-optical toggle switching, we propose to employ photo-induced magnetic anisotropy in cobalt-substituted iron garnets (YIG:Co). In these garnets, it was shown that resonant excitation of the d-d transition in Co ions can effectively change their orbital state and thus strongly affect the magnetic anisotropy[21,22]. In particular, such an excitation destroys the balance between the intrinsic cubic magneto-crystalline anisotropy and the extrinsic magnetic anisotropy promoted by the Co ions. The effec-

tive fields of the photo-induced magnetic anisotropy were found to be strong enough to permanently switch the magnetization to another bit state defined by the polarization of the laser pulse. However, toggle switching in such Co-substituted iron garnets has not been reported.

## Results and discussion
### Toggle photo-magnetic switching
We would like to note that in the previous papers, it was suggested that reported polarization-dependent switching in iron garnets could only be explained if the point group of the iron garnet films had a rather low symmetry 4. Such a low symmetry could be realized through the growth of the film on a $Gd_3Ga_5O_{12}$ (001) substrate with a miscut of 4°. It is thus proposed that growing the film on a substrate without a miscut would increase the symmetry of the point group up to 4 mm and facilitate the toggle switching. In order to test this hypothesis we grew a new Co-substituted iron garnet film on $Gd_3Ga_5O_{12}$ (001) substrate which with precision of 0.1° had no miscut (see "Methods").

Figure 1a, b shows magneto-optical images of the iron-garnet film before and after illumination with subsequent femtosecond laser pul-

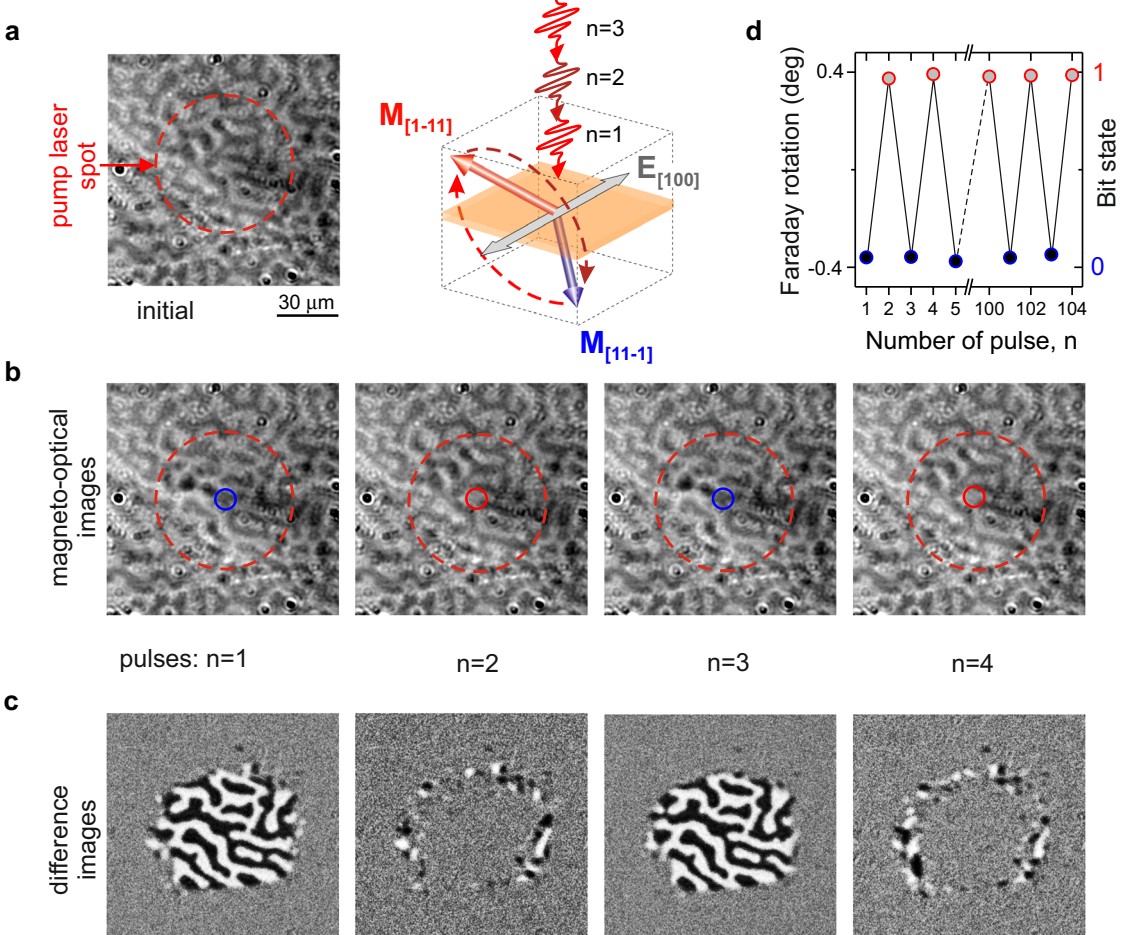

**Fig. 1 | Cold all-optical toggle switching of magnetization. a** (left panel) Magnetic domain pattern, where initially the magnetization can be either in the [11−1] state (black domain) or in the [1−11] state (white domain). Schematic illustration of the magnetic switching between [11−1] and [1−11] in iron garnet (right panel). **b** Magneto-optical images of Co-substituted yttrium iron garnet (YIG:Co) after illumination with a series of pump pulses. The fluence of a single laser pulse is 50 mJ/cm². The central wavelength of the laser is 1300 nm. The pump is linearly polarized along the [100] crystallographic axis. Dashed circles indicate the area exposed to the laser excitation. Small solid circles indicate the region, where the

measurements of the magneto-optical Faraday effect shown in (**a**) were done. **c** Differential magneto-optical images obtained as the difference between the images after $n$-th and $(n − 1)$-th pump pulses. The images reveal the toggle switching. As the switched states were captured as static images, the remaining ring on the switched spot's edge results from a domain wall motion on a much longer (microsecond) time-scale, as also observed in metallic alloys[9]. **d** Changes of the Faraday rotation in the black domain upon excitation by a sequence of laser pulses. Every single pulse changes the magneto-optical effect suggesting that the magnetization is toggled between two stable bit-states.

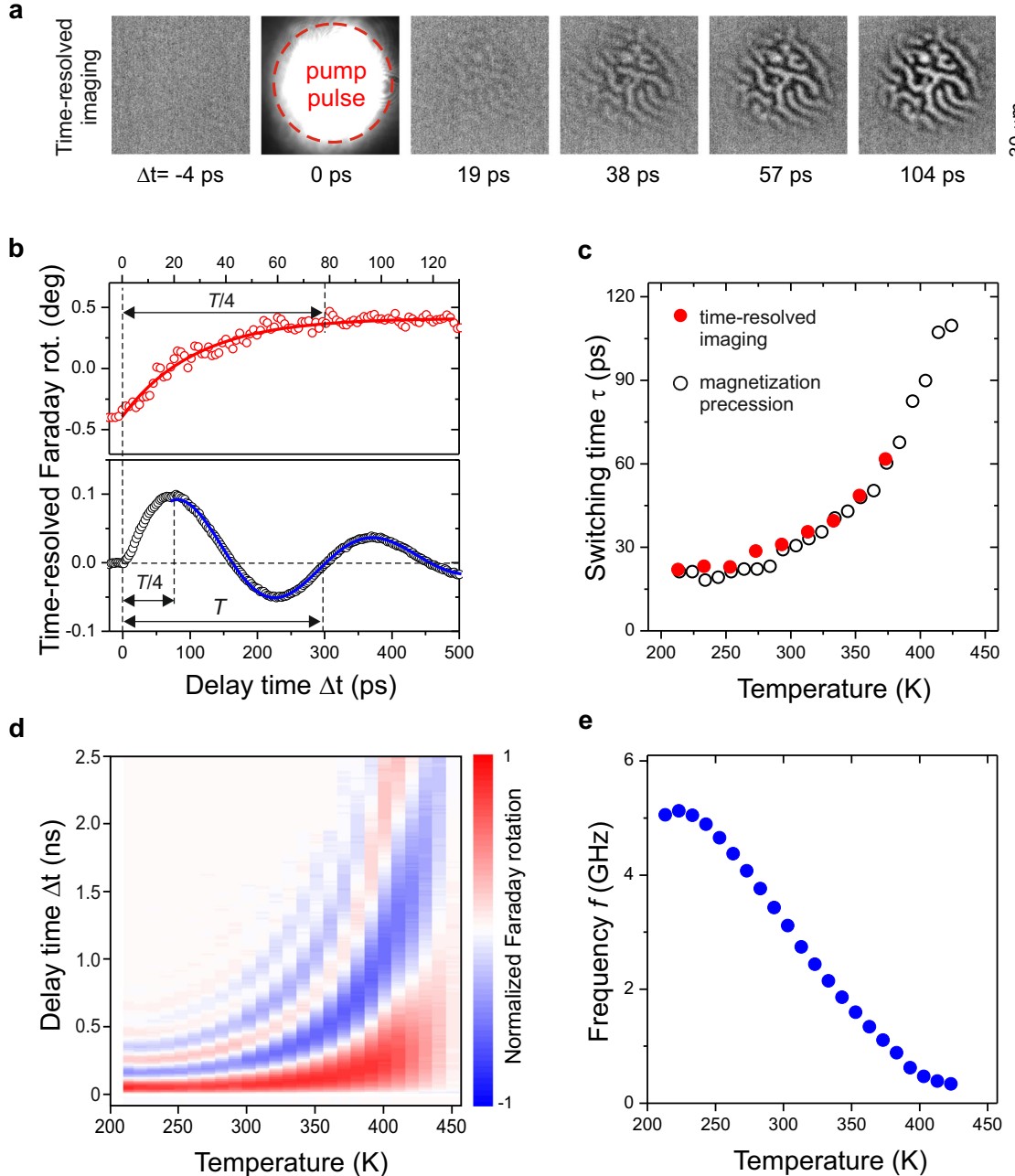

**Fig. 2 | Dynamics of the toggle switching. a** Time-resolved differential images obtained as the difference between the images after the $n$-th and $(n − 1)$-th pump pulses at room temperature. The sample was illuminated by the pump pulse linearly polarized $E \parallel [100]$ and the fluence of 50 mJ/cm². **b** (upper panel) Temporal evolution of the magnetic contrast in time-resolved imaging, and determination of the characteristic switching time $\tau$ by fitting $1−\exp(−\Delta t/\tau)$ function (red solid line) to the data. **b** (lower panel) Time-resolved Faraday effect resulting from low amplitude precession, which allows for the determination of the precession period and the frequency by fitting a damped sin function (blue solid line) to the data. **c** Switching time $\tau$ as a function of temperature deduced from time-resolved imaging (full dots) and magnetization precession (open dots) measurements. **d** The color map of the transient Faraday effect originating from the magnetization precession triggered by the pump with the fluence of 10 mJ/cm² in the range 200-450 K (see "Methods"). **e**, Precession frequency $f$ deduced from the experimental data (see Fig. 7a in "Methods" and lower panel in **b**) as a function of temperature (see "Methods").

ses. In principle, from the images, it is possible to see that each pulse changes the magneto-optical contrast, which evidences each pulse reverses the out-of-plane component of the magnetization. However, the positions of the domain walls seem to be preserved. Due to the regular labyrinth-like domain structure of the garnet, which hinders clear observation of changes, the switching is presented using differential images with respect to the initial state before laser pump excitation (see Fig. 1b). It is seen that in contrast to the earlier studies of the garnet with miscut[21], every pulse toggles the magnetization state.

Hence, the magnetic domains are thus written and erased with the same pump polarization (see Fig. 1c, d). We would like to note that toggle switching, in principle, is realized between equivalent (i.e., degenerate) equilibrium spin states. In earlier report[21], switching spins employing the effect of photo-induced magnetic anisotropy was realized in iron garnets with a miscut tilted from the [001] direction. The latter, due to deviation of the easy-axis of magnetocrystalline anisotropy from <111>-directions in the garnet (see "Methods"), leads to a difference in potential barriers for switching from a magnetization state near [11−1]

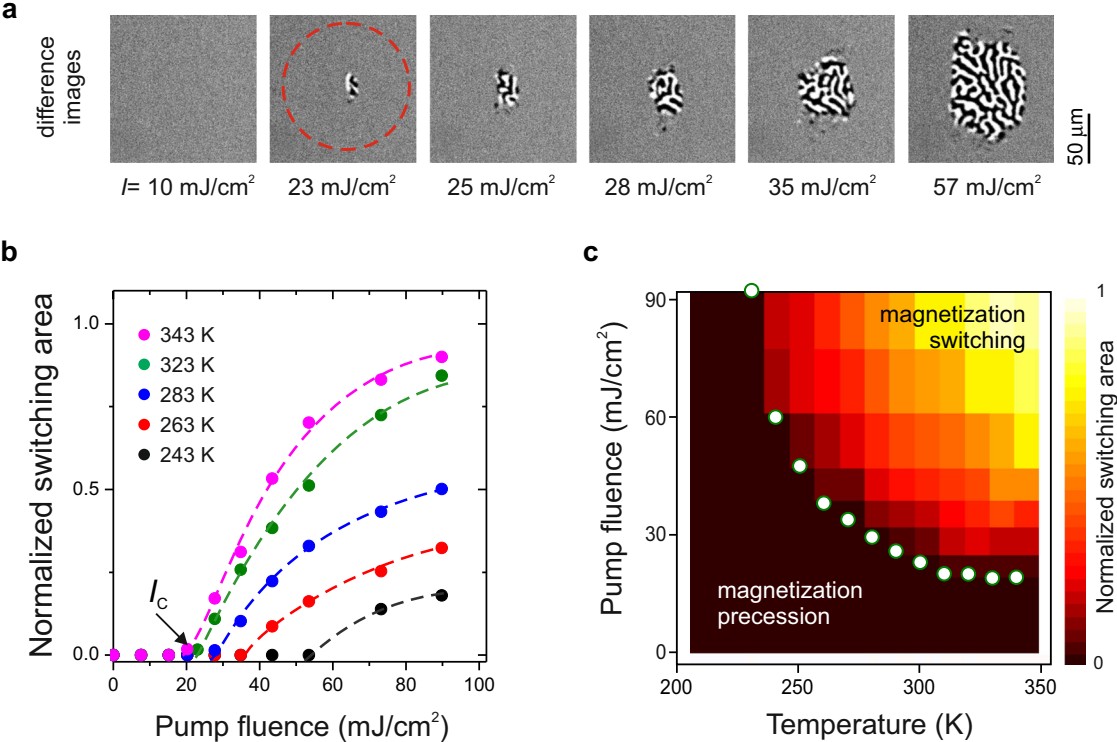

**Fig. 3 | Dissipation during all-optical toggle switching. a** The switching pattern measured at various fluences at room temperature. **b** Normalized switched area as function of fluence at different temperatures. **c** Diagram representing the switching efficiency of cold photo-magnetic recording. The color code represents the normalized switching area. The dark color corresponds to intensities insufficient for switching. The white dots highlight the minimum fluence $I_C$ required for detectable switching.

direction to a magnetization state near [1−11] direction and vice versa. Removing the miscut and thus increasing the symmetry of the film we ensure that the switching processes become equivalent. Note that regarding the used laser fluences, optical absorption, and heat capacities of the samples, the heating induced by a single laser pulse is about 1 K[21] and thus way too small to explain such dramatic magnetic changes. To the best of our knowledge, this is the first demonstration of all-optical toggle switching via a mechanism that does not rely on heat.

## Time of magnetization switching

To reveal the dynamics of the switching we performed time-resolved single-shot experiments[23] (see "Methods"). Figure 2a shows the time-resolved magneto-optical images of the YIG:Co film acquired with picosecond temporal resolution. It is seen that the contrast seen in the final image develops during 10–50 ps. We assign the characteristic time $\tau$ of the contrast developing to the switching time which is defined from the fitting an exponential increase $1-\exp(-\Delta t/\tau)$ to the data (see Fig. 2b), where $\triangle t$ is the pump-probe time delay (see "Methods"). We performed these measurements in a broad temperature range and for every temperature we deduced the switching time $\tau(T)$ (see Fig. 2c). Firstly, the switching is observed in a remarkably broad range of temperatures from 200 to 450 K and not only in the vicinity of specific points such as compensation temperature, as in the case of heat-induced toggle switching. Secondly, the switching time is strongly temperature-dependent.

In order to reveal the origin of this dependence in detail, we performed stroboscopic pump-probe measurements of the magneto-optical Faraday rotation at lower pump fluences than that required for the switching (see "Methods"). Such an excitation is not sufficient to bring the magnetization over the potential barrier and the excitation results in magnetization precession around the old thermodynamic equilibrium at the frequency of the ferromagnetic resonance[24,25]. It is

known that the precession frequency is defined by the strength of the applied magnetic field and the effective field of magnetic anisotropy[26]. Figure 2d illustrates the magnetization dynamics in the broad range of temperatures in the form of a 3D plot, where the amplitude of the detected signal is given by the color code. It is clear that the precession frequency is strongly temperature-dependent.

Using fits with a damped sine function (see Fig. 2b and "Methods") we deduced the frequency of the ferromagnetic resonance mode from the measurements in the range from 200 to 450 K and plotted the frequency $f$ as a function of temperature in Fig. 2e. It is seen that both the switching time and the frequency scale similarly with a temperature. As the external field in our measurements is zero and the magnetization precession in domains is coherent, the dependence of the frequency on temperature must originate from the temperature dependence of the effective field of magnetic anisotropy. Using Kittel's formula for the frequency of ferromagnetic resonance mode in cubic (001) crystal, we estimated the effective field of magnetic anisotropy for the frequency of precession (see Fig. 7c in "Methods").

The correlations in the temperature dependence of the switching time and the precession frequency show that the switching between the stable bit-states proceeds via magnetization precession. Obviously, the switching time between two magnetization states must be extracted from the precessional motion of magnetization during roughly $T/4$ which is similar $\tau$ obtained in time-resolved imaging (see Fig. 2b). The characteristic time $\tau$ must be also comparable with the lifetime of the photo-induced magnetic anisotropy in the Co-substitute iron garnet[24,27].

## Energy of magnetization switching

In order to estimate the intensity dissipated during the switching processes we performed measurements as a function of the pump fluence and temperature (see Fig. 3). At each temperature point, the sample was illuminated by a series of single pump pulses, maintaining

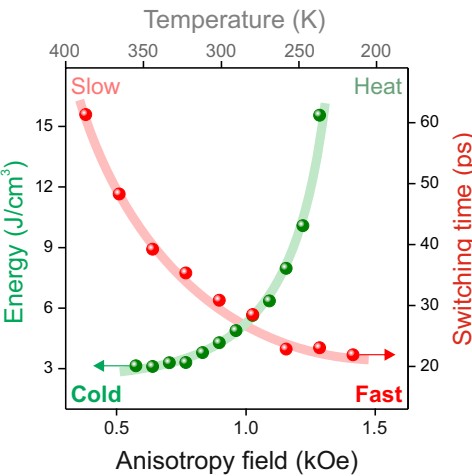

**Fig. 4 | Energy-time relation of the "cold" toggle photo-magnetic switching.** Green dots show the minimum heat load that accompanies the switching as function of temperature. Red dots show the corresponding switching time as function of temperature. Solid lines are guides for the eye.

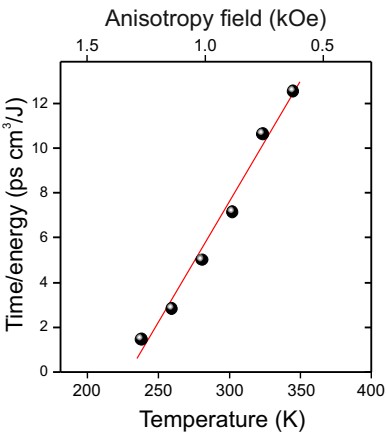

**Fig. 5 | Temperature dependence of the time-energy ratio of the switching in YIG:Co film.** The ratio between the switching time and the absorbed energy density is deduced from data shown in Fig. 4 and plotted as function of temperature. Solid line is a linear fit with a slope of $0.12 \pm 0.01$ ps•cm$^3$/J•K.

consistent focusing and fluence in the range of 1–90 mJ/cm$^2$. The switched area was defined as an envelope covering the pattern of both labyrinth-like switched domain states visible on differential images (see Fig. 3a). While reducing the fluence, we see a decrease in the switched area. Extrapolating the experimental data to the *x*-axis, we obtain an estimate of the threshold fluence $I_C$ required for the switching (see Fig. 3b). Figure 3c shows a summary of the measurements in the form of a 3D plot, where the color code indicates the normalized switched area.

The volume density of heat dissipated during toggle switching of magnetization in YIG:Co is estimated as a fraction of the threshold fluence absorbed by the sample (see "Methods"). Figure 4 shows that the switching time correlates with the dissipated energy. By choosing the sample temperature or samples with different anisotropy fields, we can either speed up the switching at the cost of larger dissipations or reduce the dissipation at the cost of the switching time. We note that the heat load during the toggle switching is found in the range from 15 to 3 J/cm$^3$, which is much smaller compared to the heat load in heat-induced toggle switching (about 1500 J/cm$^3$)[9].

We report on a new mechanism of all-optical 'cold' toggle switching in dielectric ferrimagnets, which does not rely on heat and

facilitates writing of magnetic bits with a much lower heat load than in the case of heat-induced toggle switching in metallic ferrimagnets. This new mechanism of toggle switching can be realized in a remarkably broad range of temperatures and not only in the vicinity of special points such as compensation temperature in RE-TM ferrimagnetic alloys. In particular, the ability to adjust the competition between cubic and uniaxial growth-induced anisotropy in photomagnetic materials allows for changing the angle between magnetization states, which can also reduce the switching energy threshold. The switching time in this mechanism is roughly equal to the quarter of the period of spin precession in a wide temperature range. Hence tuning the laser pulse fluence and thus strengthening the photo-induced magnetic anisotropy accelerates the switching. It is clear, however, that in this mechanism the switching time and the heat load are competing parameters, where the experimentally deduced ratio time/energy switching is increasing with temperature (see Fig. 5). We anticipate that using a train of identical laser pulses and employing the discovered switching mechanism one can toggle the magnetization of iron garnet at the frequency reaching 50 GHz. Such a high amplitude magnetization changes at so high frequency with the minimum heat load, leading to a remarkably small temperature increase of only about 0.6 K, opens up new opportunities in technologically relevant fields such as magneto-photonics[28,29], magnonics[30–32], superconducting[33] and topological spintronics[34,35] in particular. Further design of photo-sensitive ferrimagnets and antiferromagnets opens a plethora of opportunities for ultrafast and least dissipative magnetic writing with the help of light.

## Methods
### Materials
The investigated thin films of Co-doped monocrystalline yttrium iron garnet (YIG:Co) have the composition of $Y_2CaFe_{3.9}Co_{0.1}GeO_{12}$ and the thickness of 8 μm. The garnets were fabricated by liquid phase epitaxy method on the gadolinium gallium garnet ($Gd_3Ga_5O_{12}$) on (001)-plane substrate. The substrate is characterized by pure cubic symmetry without miscut angle with a precision of 0.1°. The saturation magnetization at room temperature was $4\pi M_s = 75$ G and the Néel temperature was 445 K. The magnetic anisotropy of the sample is defined at room temperature. The constant of the cubic magnetic anisotropy is $K_c = -5.5 \times 10^3$ erg/cm$^3$, and for the uniaxial anisotropy the constant is $K_u = 0.6 \times 10^3$ erg/cm$^3$. This configuration results in four distinct easy magnetization axes in <111>-directions leading to eight possible magnetization states (Fig. 6a). Pure cubic symmetry in relation to sample with a 4° miscut angle[21] leads to the regular magnetic domain structure. Temperature changes can influence the orientations of the easy magnetization axes and can induce spin-reorientation phase transitions below 170 K[36]. To the best of our knowledge, no compensation points have been reported for samples of these particular compositions so far[36,37]. In the range of temperatures 200–450 K, four easy magnetization axes along the <111> directions are maintained. The domain structures of the YIG:Co samples with non-miscut and miscut (as reference) angles are shown in Fig. 6. Notably, the image contrast corresponds to the value of the out-of-plane magnetization component [001], while the direction of the initially applied magnetic field establishes the in-plane [−110] component for both visible domains.

### Experimental technique for static and time-resolved measurements
For static (not time-resolved) measurements we conducted an experiment where the sample surface was visualized with the help of a polarization magneto-optical microscope using LED as a source of light. For the toggle-switching case, the sample was exposed by 50 fs pump pulses. The pulses were linearly polarized along the [100] crystal axis, with the central wavelength at 1300 nm. In this case, the particular wavelength was selected to resonantly pump the d-d transition in Co-ions, which provides the strongest photo-induced changes of magnetic anistropy[22,38].

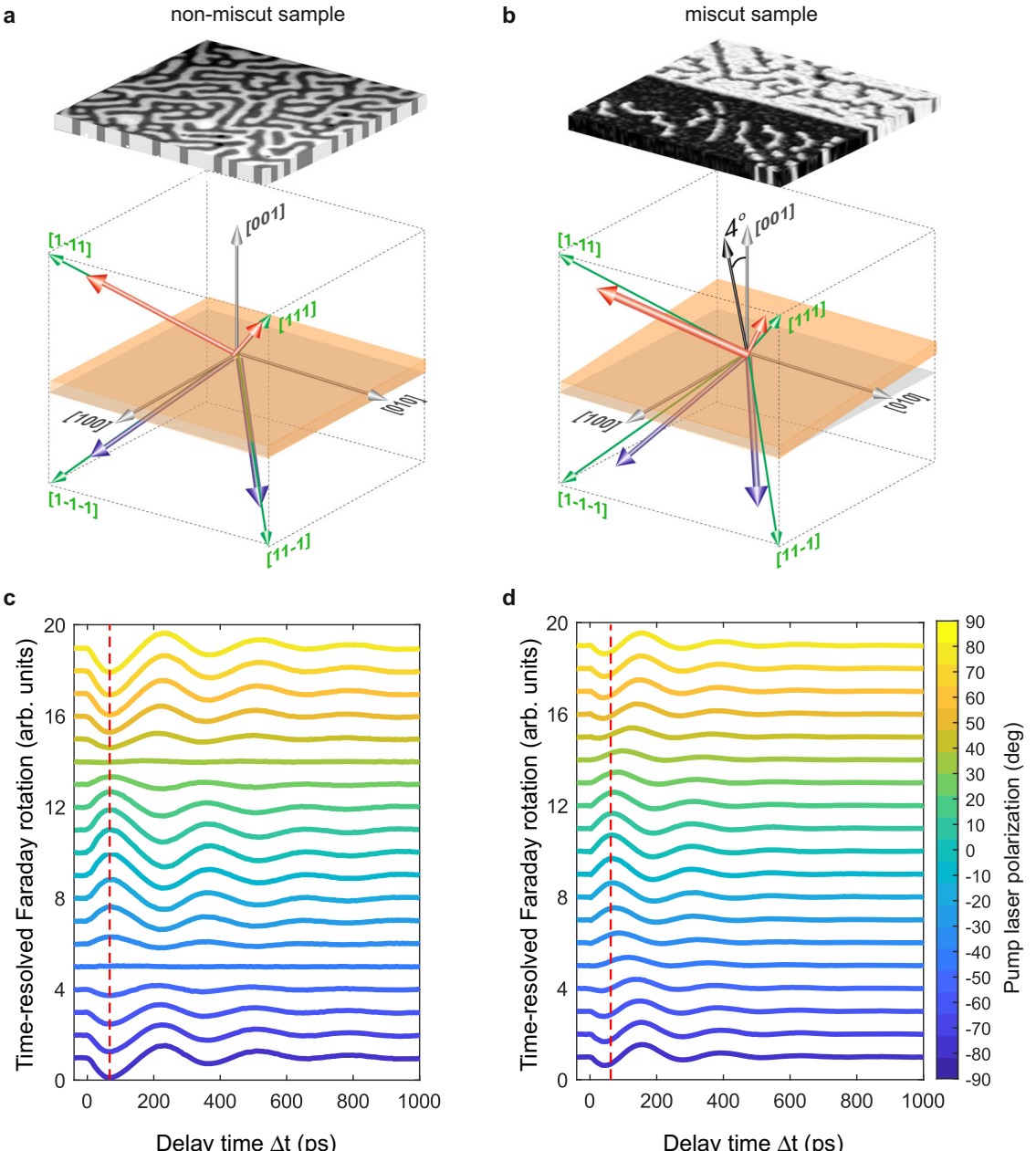

**Fig. 6 | Comparison of the YIG:Co films grown on substrates with and without miscut.** The easy magnetization axes and the magnetic domain structure in the YIG:Co at zero magnetic field for the sample without miscut are shown on (**a**), with miscut on (**b**). Laser-induced transient Faraday rotation representing laser-induced spin dynamics measured at various orientations of pump polarization (see the color code). **c** shows the measurements on the sample without miscut. **d** shows the measurements on the sample with miscut. The angle of the pump polarization set as 0° is parallel with respect to the [100] crystallographic axis in YIG:Co films. The pump fluence was set to 10 mJ/cm².

For time-resolved measurements, we employed a laser system that generated ultrashort laser pulses with a duration of 35 fs and the base repetition rate of 1 kHz. The pulses were generated by a Ti:Sapphire oscillator combined with an amplifier. The pump and probe beams were directed through separate optical parametric amplifiers, with the pump beam set to a wavelength of 1300 nm to achieve the best switching efficiency. Magnetization precession was revealed using a standard pump-probe setup. The pump beam was modulated at 500 Hz using a chopper to utilize lock-in detection. The polarization plane of the pump beam was aligned to obtain the largest precession amplitude, i.e., it was set along one of the <100>-type directions (see Fig. 6c). Subsequently, the probe beam, taken directly from the amplifier with the wavelength of 800 nm was directed onto the sample at normal incidence. After the

sample the beam was passed towards a half-wave plate and a Wollaston prism, which splits it into two linearly polarized beams with mutually perpendicular polarizations. These beams were then directed into two separate branches of a balanced photodiode bridge, which detected the difference in the intensities of light at the photodiodes. The difference scales linearly with the Faraday rotation if the latter is small. Additionally, to boost the signal-to-noise ratio, we used a boxcar integrator. The time delay $\Delta t$ was controlled by adjusting the optical path of the pump beam through a motorized translation stage. For imaging, the probe beam was deliberately defocused to cover a sufficiently large sample area. The wavelength of the probe was set to 650 nm to optimize the ratio between the Faraday rotation and absorption of light in the material. The pump beam was focused onto the sample, creating a spot

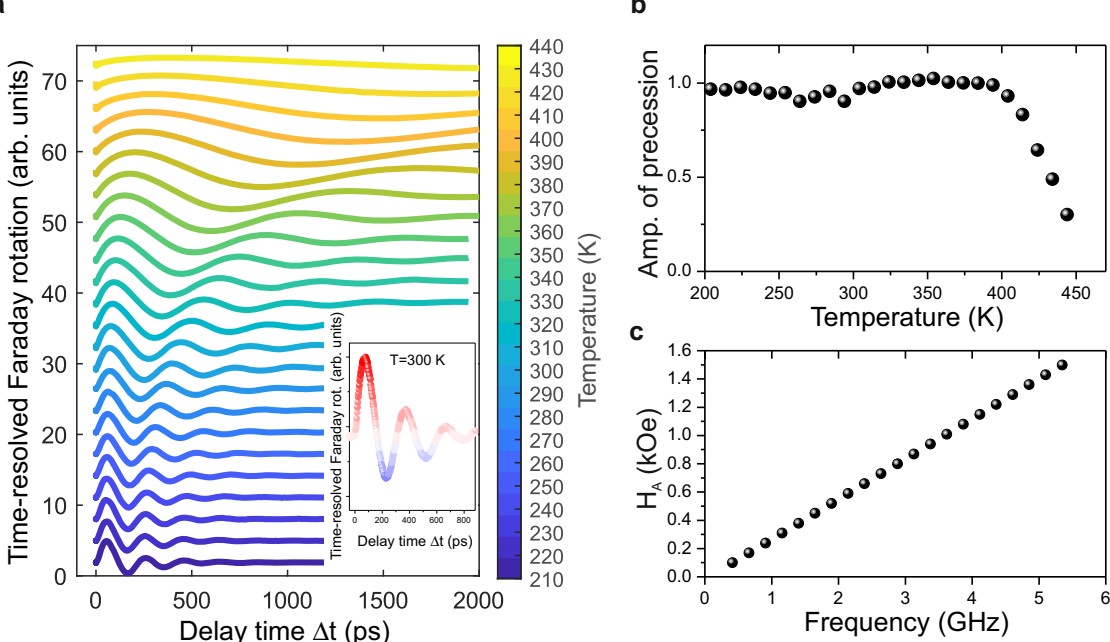

**Fig. 7 | Temperature dependence on magnetization precession in YIG:Co.**
**a** Time-resolved magnetization precession at zero magnetic field. Inset shows the magnetization precession at 300 K with a color code used in the 3D map in Fig. 2d. The precession traces show the out-of-plane magnetization component dynamics stimulated by the pump pulse with polarization $E \parallel [100]$ and fluence of 10 mJ/cm² in the pump and probe geometry. **b** Temperature dependence of the normalized amplitude of the photoinduced magnetization precession which is extracted from time-resolved Faraday rotation. **c** The results of the calculation of effective anisotropy field $H_A$ as a function of frequency $f$ using the Kittel formula[26].

with a diameter of approximately 120 μm and a fluence $I_c$ of 50 mJ/cm². The absorbed heat load in the volume required for obtaining switching was estimated as $q = a_\lambda I_c / d$, where $a_\lambda$ is the absorption coefficient which was measured at the wavelength of 1300 nm ($a_\lambda = 0.12$) and $d$ is its thickness. All measurements were conducted without applying an external magnetic field.

The data collection involved creating a series of time-resolved magneto-optical images, illustrating the spatial and temporal changes in magnetization[23]. For each image in the stack, we followed the sequence as follows: capturing a background image (illuminated solely by the probe beam), and capturing a dynamic image (both pump and probe beams illuminating the sample). To improve the signal-to-noise ratio and isolate the changes induced by the pump, we subtracted the background image from each dynamic pump-probe image, resulting in a stack of differential images (see Fig. 2a). This relative change allowed us to observe the time-dependent evolution of the magnetization projection $\Delta M_z$ along the [001] axis, which could be normalized to the static Faraday rotation.

The pure cubic symmetry maintains itself in the polarization dependence. The lack of miscut in contrast to miscut-sample allows for complete suppression of the precession signal for the case of the orthogonal arrangement of a linearly polarized pulse $E \parallel [110]/[1\text{–}10]$ (orientation pump polarization state is $\varphi = 45°/\text{–}45°$ to the crystallographic axis (Fig. 6c, d).

The temperature dependence of magnetization dynamics, which was utilized to create the color map in Fig. 2d, is presented in Fig. 7a in a waterfall plot. The color decoding is provided in Fig. 2d. The presented traces were normalized to adapt to the color code and fitted with a damped sine function with frequency $f$ in the form of: $\Delta\theta_F(\Delta t) = A \sin(2\pi f \Delta t + \varphi) \exp(-\Delta t/\tau)$. The amplitude of precession ($A$) dependence on temperature is shown in Fig. 7b, which correlates with the dependence of magnetization saturation $M_s$ in the sample[39]. In order to calibrate the effective field of magnetic anisotropy $H_A$ (see Fig. 7c), we calculated the precession frequency using the

Kittel's formula[26] $f = \frac{\gamma}{2\pi M_s \sin\theta} \sqrt{\frac{\delta^2 W}{\delta\theta^2}\frac{\delta^2 W}{\delta\varphi^2} - \left(\frac{\delta^2 W}{\delta\theta\delta\varphi}\right)^2}$, where the $\gamma$ is the gyromagnetic ratio and $\theta$ and $\varphi$ are the polar and azimuthal angles, respectively. The energy was defined as: $W(\theta,\varphi) = K_c(\sin^4\theta\sin^2\varphi\cos^2\theta + \sin^2\theta\cos^2\theta\cos^2\varphi + \sin^2\theta\cos^2\theta\sin^2\varphi) + K_u\sin^2\theta - 2\pi M_s^2\sin^2\theta$.

## Data availability
The data that support the findings of this study are available from the corresponding author upon reasonable request.

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

## Acknowledgements

We acknowledge support from the grant of the Foundation for Polish Science POIR.04.04.00-00-413C/17-00, the European Union's Horizon 2020 Research and Innovation Programme under the Marie Skłodowska-Curie grant agreement No 861300 (COMRAD) and the European Research Council ERC grant agreement No 101054664 (SPARTACUS). We thank Professor A. Kirilyuk for the fruitful discussions.

## Author contributions

A.S. conceived the project with contributions from A.V.K. The measurements were performed by T.Z. All the authors discussed the results. A.V.K. and A.S. co-wrote the manuscript with contributions from T.Z. The project was coordinated by A.S.

## Competing interests

The authors declare no competing interests.
