## [Peer Review File · Nature Communications]

Reviewers' Comments:

Reviewer #1:

Remarks to the Author:

In this work the authors use femtosecond laser pulses to switch magnetic domains in Co doped yttrium iron garnet. The switching is a toggle switching whereby each pulse causes a switch to a different energy minimum, but without an external field or external directional bias to break the symmetry between the states. The authors explain their results by a sudden increase in uniaxial anisotropy due to a d-d transition in the Co. This perturbation stimulates a larger precession of the magnetisation which, if large enough, can lead to the magnetisation crossing the energy barrier between the cubic ground states. The timing is important to ensure that the magnetisation does not recross the barrier, but overall the behaviour is deterministic. The authors are careful to perform experiments to support their interpretation, especially the results in Fig. 2 which are very persuasive. One of the most important claims in the work is that the switching occurs with very little heating, about two orders of magnitude lower than previous laser induced toggle switching mechanisms. Again, supporting experiments are performed and the analysis is persuasive.

Overall I found the work to be very interesting and worthy of publication in Nature Communications. Toggle switching without putting a lot of heat into the system is of significant interest in this field.

I have the following comments which should be addressed by the authors before publication:

- The sample used is not simply Co doped YIG but contains both Ca and Ge. Importantly the Fe is about 20% substituted by Ge and the Neel temperature is about 100K lower than bulk YIG. Highly substituted YIG such as this has quite different magnetic properties to pure YIG, with the potential for reorientation transitions and compensation points. The authors state that there is no reorientation transition within the range of temperatures they measure and cite one of their earlier works [J. Magn. Magn. Mater. 254-255, 562 (2003)]. However, in that work the full composition of the sample is not given. Can the authors confirm that the samples are either the same, or that they have measured the magnetisation properties of the sample used in the present study. Ideally they should present data characterising this sample in a supplementary information. The presence of a compensation point or reorientation transition would seriously affect the interpretation of their results.

- The toggle switching shown here is reasonably straight forward to understand from the precessional dynamics of the system and I don't think the discussion about Curie's principle are useful. The dynamics of the precessional toggle switching here is not significantly different from the examples cited in the manuscript. Moreover, there is an asymmetry in the system which is that the magnetisation always precesses in a counter clockwise direction. I believe this asymmetry, combined with precise timing is what enables deterministic toggle switching. This does not appear to be a violation of Curie's principle as there is an inherent asymmetry in the system. If we consider instead a hypothetical system where magnetisation does not have precessional dynamics of a single chirality but simply diffuses within the minimum of the energy well, then there is no asymmetry in the system or the impulse and a toggle switching would be unexpected and appear to violate Curie's principle. This example is perhaps closer to the situation with GdFeCo where the high temperature and small magnetisation makes the precessional dynamics less relevant and the toggle switching more surprising. I do not insist that all references are removed to Curie's principle, but do not think it is necessarily violated here and strong claims should not be made on this point. The scientific results in the work are of interest even without invoking Curie's principle.

- Throughout the manuscript the substrate is referred to with the chemical formula $Gd_3Fe_5O_{12}$ (a ferrimagnet with a compensation point), but I'm sure it's incorrect. In the methods section it is named as gadolinium gallium garnet, which is the normal paramagnetic substrate for garnets. Probably the chemical formula needs fixing throughout.

- In Fig. 4 the left axis has a typo "Enegrty" -> "Energy"

Reviewer #2:

Remarks to the Author:

The authors have reported all optical toggle switching in Co doped YIG layer. YIG being a ferrimagnetic insulator, the reported toggle switching of magnetization is different from that reported earlier for metallic ferrimagnets, where ultrafast heating played a major role in the switching phenomena. In the case of Co doped YIG, the occurrence of a photo induced magnetic anisotropy on a sub-precessional timescale is responsible for the toggle switching, which shows a new pathway of all optical switching of magnetization on 10s of picosecond timescale.

I have a few doubts :

1) In earlier reports on polarization based all optical switching in YIG, the phenomena was attributed to photo induced magnetic anisotropy. It is understandable that an occurrence of transient magnetic anisotropy can lead to polarization dependent magnetic switching. But, for toggling, this photo-induced anisotropy has to change direction with every pulse, which is not the case. In this respect, what is the reason for toggle switching which was not present earlier? Some argument based on miscut of substrate is given. However, it is not clear how an increase in the symmetry of orbitals will lead to toggling.

2) Page 2 line 74: I think it should be 'Figure 1a-b', instead of 'Figure 1a'.

3) In Fig. 1c, with the application of even no. of laser pulses, the switched spot is switching, leaving a ring. Why is the ring not vanishing?

4) In my understanding, the laser induced anisotropy should be dependent on laser fluence, not on temperature. As shown in Fig. 2 and 3, the switching time increases with temperature and the threshold fluence decreases with temperature. From these relations, is it possible draw any connection between switching time and threshold fluence in terms of magnetic precession time?

5) In case of this insulator, how have you calculated the dissipated heat? The explanation of Fig. 4 is not very clear to me.

REPLY TO REVIEWERS

Reviewer A

1. *The sample used is not simply Co doped YIG but contains both Ca and Ge. Importantly the Fe is about 20% substituted by Ge and the Neel temperature is about 100K lower than bulk YIG. Highly substituted YIG such as this has quite different magnetic properties to pure YIG, with the potential for reorientation transitions and compensation points. The authors state that there is no reorientation transition within the range of temperatures they measure and cite one of their earlier works [J. Magn. Magn. Mater. 254-255, 562 (2003)]. However, in that work the full composition of the sample is not given. Can the authors confirm that the samples are either the same, or that they have measured the magnetisation properties of the sample used in the present study. Ideally they should present data characterising this sample in a supplementary information. The presence of a compensation point or reorientation transition would seriously affect the interpretation of their results.*

Response: We thank the Referee for this comment. We do confirm that the samples studied in the present manuscript have the same composition as those mentioned in the reference [J. Magn. Magn. Mater. 254-255, 562 (2003)]. To response to the criticism, we have included a brief clarification along with an additional reference [39] providing more info about the composition of the samples in our experiments. In particular, concentrations of Co and Ca-Ge were determined with the help of electron probe microanalysis. To the best of our knowledge, no compensation points have been reported for samples of these particular compositions so far.

In the present version the Methods section states: "Temperature changes can influence the orientations of the easy magnetization axes and can induce spin-reorientation phase transitions **below 170 K³⁸. To the best of our knowledge, no compensation points have been reported for samples of these particular compositions so far³⁸⁻³⁹.**"

[39] R. Jabłoński, A. Maziewski, M. Tekielak, and J. M. Desvignes, FMR Study of Co-Substituted Yttrium Iron Garnet Films, J. Magn. Magn. Mater. 160, 367 (1996).

2. *The toggle switching shown here is reasonably straight forward to understand from the precessional dynamics of the system and I don't think the discussion about Curie's principle are useful. The dynamics of the precessional toggle switching here is not significantly different from the examples cited in the manuscript. Moreover, there is an asymmetry in the system which is that the magnetisation always precesses in a counter clockwise direction. I believe this asymmetry, combined with precise timing is what enables deterministic toggle switching. This does not appear to be a violation of Curie's principle as there is an inherent asymmetry in the system. If we consider instead a hypothetical system where magnetisation does not have precessional dynamics of a single chirality but simply diffuses within the minimum of the energy well, then there is no asymmetry in the system or the impulse and a toggle switching would be unexpected and appear to violate Curie's principle. This example is perhaps closer to the situation with GdFeCo where the high temperature and small magnetisation makes the precessional dynamics less relevant and the toggle switching more surprising. I do not insist that all references are removed to Curie's principle, but do not think it is necessarily violated here and strong claims should not be made on this point. The scientific results in the work are of interest even without invoking Curie's principle.*

Response: We understand the concern of the Referee and fully agree that the observation reported in the paper can be explained in terms of precessional switching. We agree that stating that the observations reported in the manuscript violate Curie's principle may be too strong. At the same time, we also appreciate that the Referee does not insist that we must remove references to Curie's principle completely. We believe that the mismatch between symmetries of the cause and the response is one of the main reasons why non-specialists would see the phenomenon of toggle switching as counter-intuitive.

In order to respond to the criticism we have softened the claims about violation of Curie's principle.

We have changed the abstract: "Here we show that the regime of ultrafast toggle switching ~~beyond Curie's principle~~ can be also realized via a mechanism without relying on any heat. ~~The obvious mismatch between symmetries of the cause and the response can thus be seen as a violation of Curie's principle.~~"

We have also changed the main text: Thermodynamic Curie's principle in this case is ~~violated no longer applicable and a match of cause-response symmetries is no longer required~~, because the field is applied on a time scale, which is much shorter than the time required for the magnetization to arrive at the thermodynamic equilibrium i.e. become parallel with respect to the applied magnetic field.

3. Throughout the manuscript the substrate is referred to with the chemical formula $Gd_3Fe_5O_{12}$ (a ferrimagnet with a compensation point), but I'm sure it's incorrect. In the methods section it is named as gadolinium gallium garnet, which is the normal paramagnetic substrate for garnets. Probably the chemical formula needs fixing throughout.

Response: We thank the Referee for noticing this annoying mistake, which indeed propagated throughout the entire manuscript. In response to the criticism, everywhere in the manuscript we now use the proper chemical composition of gadolinium-gallium garnet (GGG) $Gd_3Ga_5O_{12}$.

4. In Fig. 4 the left axis has a typo "Enegrty" -> "Energy"

Response: We fixed the mentioned typo.

Reviewer B

1. In earlier reports on polarization based all optical switching in YIG, the phenomena was attributed to photo induced magnetic anisotropy. It is understandable that an occurrence of transient magnetic anisotropy can lead to polarization dependent magnetic switching. But, for toggling, this photo-induced anisotropy has to change direction with every pulse, which is not the case. In this respect, what is the reason for toggle switching which was not present earlier? Some argument based on miscut of substrate is given. However, it is not clear how an increase in the symmetry of orbitals will lead to toggling.

We thank the Referee for this remark. The previously mentioned miscut introduces degeneracy between domain states, slightly modifying the energy required to switch between particular states. In other words, the efficiencies of switching via one polarization (parallel to [100]) and the orthogonal one (parallel to [010]) were different, resulting not only in a change in torque direction but also in non-equivalence in trajectories.

To clarify, we have added in the main text: “We would like to note that toggle switching, in principle, is realized between equivalent (i.e. degenerate) equilibrium spin states. In earlier report²³, switching spins employing the effect of photo-induced magnetic anisotropy was realized in iron garnets with a miscut tilted from the [001] direction. The latter, due to deviation of the easy-axis of magnetocrystalline anisotropy from <111>-directions in the garnet (see Methods), leads to a difference in potential barriers for switching from a magnetization state near [11-1] direction to a magnetization state near [1-11] direction and vice versa. Removing the miscut and thus increasing the symmetry of the film we ensure that the switching processes become equivalent.”

2. Page 2 line 74: I think it should be ‘Figure 1a-b’, instead of ‘Figure 1a’.

We have added the mentioned remark.

3. In Fig. 1c, with the application of even no. of laser pulses, the switched spot is switching, leaving a ring. Why is the ring not vanishing?

We thank the Refree for bringing this to our attention.

We have now added a description to Figure 1: “As the switched states were captured as static images, the remaining ring on the switched spot's edge results from a domain wall motion on a much longer (microsecond) time-scale, as also observed in metallic alloys².”

4. In my understanding, the laser induced anisotropy should be dependent on laser fluence, not on temperature. As shown in Fig. 2 and 3, the switching time increases with temperature and the threshold fluence decreases with temperature. From these relations, is it possible draw any connection between switching time and threshold fluence in terms of magnetic precession time?

Response: The switching is a result of the interplay between effective fields of mutually competing magnetic anisotropies (photo-induced, magnetocrystalline, and grown-induced). While the photo-induced anisotropy is a function of laser fluence, the magnetocrystalline anisotropy is temperature dependent. We note that the switching time is well correlated with

the quarter of the period of the spin precession in a wide temperature range (see Fig. 2c). We agree with the Referee that it is indeed interesting to establish a connection between the switching time and the threshold fluence (absorbed energy density during the switching). The experimentally deduced ratio between the switching time and the energy is a linear function of temperature in the range 230-340 K with the slope of about 0.12 ps·cm³/J·K (see new Fig. 7).

To respond to this comment, we added to the main text of the manuscript the following changes in Conclusion:

Old text: “The switching time in this mechanism can be controlled by tuning the intensity of the laser pulse, but the switching time and the heat load are found to be competing parameters.”

New text: “The switching time in this mechanism is roughly equal to the quarter of the period of spin precession in a wide temperature range. Hence tuning the laser pulse fluence and thus strengthening the photo-induced magnetic anisotropy accelerates the switching. It is clear, however, that in this mechanism the switching time and the heat load are competing parameters, where the experimentally deduced ratio time/energy switching is increasing with temperature (see Fig. 7).

We have added the new Fig. 7 in Methods with dependence of time/energy ratio on the temperature and the anisotropy field.

5. In case of this insulator, how have you calculated the dissipated heat? The explanation of Fig. 4 is not very clear to me.

We thank the Referee for this remark. We have added the following information in the Methods (lines 258-260): “The absorbed heat load in the volume required for obtaining switching was estimated as $q = a_\lambda I_c/d$, where a_λ is the absorption coefficient which was measured at the wavelength of 1300 nm ($a_\lambda = 0.12$) and d is the thickness.”

Reviewers' Comments:

Reviewer #1:

Remarks to the Author:

The authors have addressed the comments from myself and the other referee well. I see no further issues that need revising and recommend publication as is.

Reviewer #2:

Remarks to the Author:

The manuscript looks fine now. I recommend it for publication.